# Seeing the Forest and the Trees: A Scoping Review of Empirical Research on Work-Life Balance

**Ka Po Wong** [1,*], **Pei-Lee Teh** [2] **and Alan Hoi Shou Chan** [3]

1   Centre for Smart Health, School of Nursing, The Hong Kong Polytechnic University, Hong Kong, China
2   School of Business, Gerontechnology Laboratory, Monash University Malaysia,
    Bandar Sunway 47500, Malaysia
3   Department of Advanced Design and Systems Engineering, City University of Hong Kong, Kowloon,
    Hong Kong, China
*   Correspondence: portia.wong@polyu.edu.hk

**Abstract:** Work–life balance (WLB), which has become a central issue in workers' everyday lives, is a global issue with a growing body of investigation into its meaning and the construction of suitable measurement scales, but varying meanings for WLB have been observed in studies. Due to these discrepancies, review or summary work is needed to identify the trends and development of WLB among workers, including (a) the commonly used WLB scales, (b) the antecedents and outcomes related to WLB and (c) the frequency of the emergence of these antecedents and outcomes. This review aims to provide an overview of empirical studies investigating the antecedents and outcomes of WLB. A total of 99 published articles from 77 journals over the period of 2006–2020 were extracted. The research methods, analysis methods, countries investigated, pivot of WLB scales used, and thematic topics and research gaps were identified. The trends of WLB, including the establishment of standard working hours, the availability of working from home, the effects of technologies on achieving WLB and the benefits of WLB for subjective wellbeing, are discussed. The research insights will provide the research directions for constructing WLB scales and investigating issues that significantly affect the WLB of employees.

**Keywords:** work–life balance; empirical research; scoping review; standard working hours; subjective wellbeing; working from home; technologies

## 1. Introduction

Sustainable work–life balance (WLB) plays a vital role in workplace health promotion and maintaining the physical and psychological wellbeing of people at work in the long term. Sustainable work–life balance has been used to describe the combined endeavour of society, organisations and employees because governments and organisations need to formulate strategies and measures to promote WLB for employees, and employers and employees negotiate directly to jointly contemplate and adopt the most suitable WLB employment practices that are in the best interests of the enterprise and employees [1,2]. Companies have promoted various WLB initiatives, such as offering flexible scheduling, creating flexible leave policies and providing educational support, to improve employees' quality of life. Still, many workers continuously lament about poor WLB.

Over the years, the relationship between work and personal life has been investigated by many researchers. Work–family conflict and work–family enrichment are the initial foci amongst scholars [3–5]. However, many researchers have argued that the family does not dominate the nonwork domain [6,7]. That is, family is not the only non-work domain of a person's life, as there are other things, such as self-care, leisure, friends and so on [8,9]. Thus, WLB has been suggested as a solution to this problem, where there is no dominance between each area of life, and it has become the primary focus in society. Balance here refers to the stability of the physical body and mindset [10]. WLB refers to achieving a

concordance between work and nonwork domains and the capability of an individual to meet the commitments of both work and nonwork activities [11]. Several reviews have been conducted to synthesise the definitions, theoretical approaches and cause-and-effect of WLB [12–14].

Further, the selected studies of some reviews for WLB have defined conflict between work and personal life the same as WLB [15–17], which cannot be regarded as a review of WLB, while parts of the review are still on point. To our knowledge, there are very few scoping reviews reported that analyse the empirical studies of WLB alone. Most reviews related to WLB include conflict and enrichment, which may not effectively reflect the actual causality of WLB because balance is conceptually different from conflict and enrichment. Conflict and enrichment include one domain's characteristics that affect or shape another domain, whereas balance has not yet emphasised it. While conflict and enrichment are linkage mechanisms between work and family, WLB reflects summative characteristics of individuals engaging in and enjoying multiple roles in the work and personal life spheres [18]. Balance is a more holistic perspective than the experiences of conflict and enrichment of an individual [18]. The emergence of conflict and enrichment is related to the expected behaviours of employees; nevertheless, WLB is to manage the conflicts between work and life roles, which is conceptually more demanding than conflict and enrichment. Figure 1 illustrates how conflict and enrichment exist in the state of WLB. The overlap between work and life domains indicates conflict and enrichment. WLB is a dynamic state, and the interaction between work and life domains causes conflict or enrichment. Conflict or enrichment may vary over time and is only part of WLB. A review that only includes studies investigating WLB is needed to deepen our understanding of WLB and to avoid focusing too much on conflict and enrichment when evaluating WLB. Furthermore, the body of research on WLB has greatly increased. There have been reviews on the investigation of the meaning and concept of WLB [13,14], while these reviews consider conflict and enrichment between work and family/life domains to be similar to WLB, and the taxonomy of the factors and outcomes is in-depth. That is, predictors of WLB have been classified into personal and organisational levels, and outcomes of WLB have been categorised into work-, nonwork- and stress-related perspectives. The lack of a consistent definition of WLB and the focus on the work–family interface has reduced the concept's importance to research and practice.

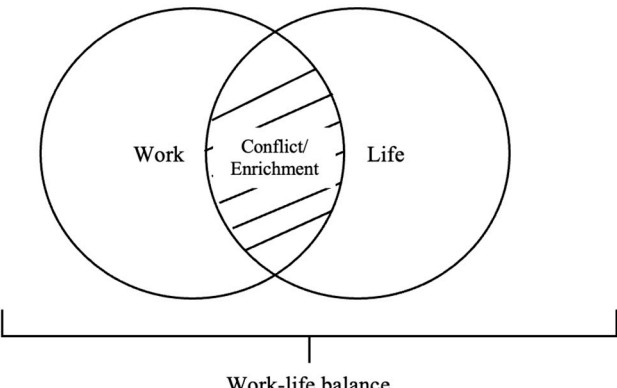

**Figure 1.** The emergence of role conflict and balance in WLB state.

In this scoping review, to enhance the prominence and uniqueness of WLB for future research, only empirical studies that investigated WLB were included, while those employing conflict and enrichment as constructs were excluded. The trends and development of WLB have been updated through previous empirical studies. We explored the commonly used WLB scales that focus only on the cause-and-effect related to WLB, determined the pivots of the WLB scales, distinguished the antecedents and outcomes of WLB and identified the main thematic topics from the antecedents and outcomes of WLB. By achieving

these goals, it is hoped that the findings can provide useful insights for future WLB scale development and further studies regarding the causes, effects and mechanisms of WLB.

This review contributes to the existing literature in several ways. First, this study offers a summary of the foci of the WLB scales used to enhance our understanding of how researchers comprehend WLB, as reflected in their research works. Second, this study sheds light on the relationships between WLB and various antecedents and outcomes identified from previous empirical studies and categorises the taxonomy of these antecedents and outcomes into different thematic topics. More specifically, the taxonomy provides further insight into the 15-year development and foci on WLB amongst researchers. Third, this study proposes a future agenda of the trending topics and controversial issues related to WLB development.

*History and Theory of WLB*

WLB was at the cutting edge of research interest between the 1980s and 1990s because of the influx of a female workforce [19]. The culture of long working hours and unpaid overtime prevailed in the 1950s, as different innovative technologies were developed, through which countries became industrialised [20]. The high work demands put tremendous pressure on workers, and demand grew for shortening working hours and creating more time for family. In response, working hours for women and children were limited, and five working days per week was promoted. After this time, a standard of forty-four working hours per week was established in the United States [21]. However, workers in some Asian countries or regions still suffer from long working hours [22,23]. At that time, research concerning the impacts of work on the family was the focus, and the influences of work on other aspects were not considered. Overall, the focus of the research was narrow.

Life is composed of various segments, including family, work, leisure and social life [24,25]. Sieber [25] and Marks [26] indicated that balance is entirely devoting oneself to each role. The mutually influencing relationship between the family realm and work was developed by Moen and Kanter [27]. Role balance theory, which suggested that workers sought meaningful experiences in both work and family roles, was proposed by Marks and MacDermid [28]. Pleck [29] claimed that women's work experiences spill over to the family and that men's experiences of family spill over to work. The spillover theory was consolidated by Staines [30], who argued that the spillover from both sides could be either positive or negative. This work complemented the compensation theory, namely, that the negative experiences of one side can be compensated for by the positive experiences of the other side. Greenhaus and Beutell [31] proposed a new orientation, namely, that balance can be achieved by minimising the conflict in each role. However, high involvement in one domain leads to a scarcity of time dealing with the demands of the other role, which causes stress.

Subsequently, boundary theory proposed that each individual must manage their work and personal lives through the integration and segmentation of each role [32,33]. Integration means that no physical or mental boundary exists between work life and non-work life, and thus spillover occurs. Segmentation is contrary to integration in that a clear borderline divides each domain of life, and there are no influences between the segments. Subsequently, some researchers have proposed that the attainment of satisfactory experiences amongst all life domains is WLB [34,35]. The definition of WLB was vigorously contested. Frone [36] stated "work-family balance is a lack of conflict or interference between work and family roles". Greenhaus et al. [37] defined WLB as equivalence in time, involvement and contentment across all segments of life. Lockwood [19] suggested that WLB is an equivalent demand from work and nonwork domains. A concordance between the needs and resources in the work and family roles is further proposed by Voydanoff [38] as the definition of WLB. Grzywacz and Carlson [39] defined WLB as the accomplishment of role expectations. It can be seen that the concepts of WLB proposed in previous studies are mainly satisfaction in work and life domains; no interference between work and life spheres; and equal time, demands and satisfaction, and fulfilment of role

expectations. These concepts of WLB are clearly distinct from conflict and enrichment. Subsequently, many researchers have attempted to define WLB, but they differ only on minor points [40,41], and thus, there is still no consensus as to the definition of WLB. Various theories have been proposed to express the meaning of WLB (e.g., [28,30,33]); however, the definitions related to WLB remained undecided.

In the last ten years, the WLB literature emphasises the perceived involvement, autonomy, supports and meaningfulness of different roles [42–47]. The antecedents of WLB, such as working hours and organisational support, continue to expand in line with the WLB policies formulated by organisations and the impacts of technology [48–50]. For instance, the more support a person has from an organisation, the easier it is to achieve better WLB [48]. Organisational support can manifest itself as encouragement and assistance to the employees. Workers may spend less time with their families due to longer working hours, and the reduction in family time may lead to poor WLB [50]. Due to the increasing pressure of everyday life, workers are prone to have poor WLB. Poor WLB may result in a decline in health, for example, as workers tend to sleep less and increase their stress levels [49,50]. Reduced sleep time and increased stress levels can weaken the immune system and exacerbate disease symptoms [50]. WLB makes employees feel satisfied at work because workers with WLB implied that they do not need to be bothered by problems outside of work, and they can work well during work hours. Hence, the perceived outcomes of WLB are also expanding as the influences focus not only on satisfaction with life and work, feelings and performances at different roles, and physical and mental health, but on creativity and financial wellbeing, as well. Meanwhile, considering that the recent WLB-related review conducted by Sirgy and Lee [14] considered conflict and enrichment as WLB and focused on work–family balance instead of WLB, it is essential to synthesise and update the research associated with WLB. This study aims to: (1) characterise the participants and study characteristics of empirical studies; (2) identify the WLB scales used in the empirical studies and categorise the WLB scales based on their natures; (3) identify the antecedents and outcomes related to WLB and categorise them into different thematic topics; (4) summarise the correlations among WLB, its antecedents and outcomes and the frequency of these correlations in three different periods, including 2006–2010, 2011–2015 and 2016–2020; (5) highlight the potential trend of WLB development; and (6) suggest research gaps and avenues for future research directions.

## 2. Materials and Methods

### 2.1. Search Strategy

To identify peer-reviewed articles on the measures or models of the relationship between the work and life domains, several databases—Web of Science, PsycINFO, ProQuest, Scopus and Google Scholar—were used for eligible studies published up to 31 March 2020. This Scoping review was registered in the PROSPERO International Prospective Register of Systematic Reviews (REGISTRATION NUMBER: CRD CRD42022363550). Two reviewers (K.P.W. and P.-L.T.) extracted the potentially qualified papers by searching the keywords following the PICO principle [51,52].

Population: 'workers' OR 'employees' OR 'labour' OR 'labor' OR 'workforce' OR 'white collar' OR 'blue collar'
Intervention: 'work–life' OR 'work/life' OR 'work–personal life' OR 'work/personal life' OR 'work/non-work' OR 'work–family' OR 'work/family' AND 'balance'
Outcomes: 'job satisfaction' OR 'organisational support' OR 'turnover intentions' OR 'stress' OR 'organisational commitment' OR 'job performance' OR 'supervisor support' OR 'job involvement' OR 'work engagement' OR 'work motivation' OR 'innovation' OR 'organisational citizenship behaviour' OR 'quality' OR 'cooperation' OR 'productivity' OR 'leadership' OR 'loyalty' OR 'commitment' OR 'recognition' OR ' wellbeing' OR 'burnout' OR 'life satisfaction' OR 'anxiety' OR 'depression' OR 'life orientation' or 'life quality' OR 'life meaningfulness' OR 'passion' OR 'flexible' OR 'work leave' OR 'childcare support' OR 'job sharing' OR 'autonomy' OR 'meaningfulness' OR 'family satisfaction' OR 'dependent

care responsibilities' OR 'family performance' OR 'financial wellbeing' OR 'working hours' OR 'career opportunities' OR 'resources'

## 2.2. Eligibility

The titles and abstracts of the articles were screened by two reviewers (K.P.W. and P.-L.T.) according to the study inclusion criteria: (a) working adults aged 18 or above; (b) empirical studies that provide correlations between WLB and its antecedents and outcomes; (c) articles published between 2006 and 2020; (d) articles published in English. In this stage, 1051 articles were collected. Exclusion criteria included: (a) studies investigating conflict, enhancement, enrichment, facilitation, interference and positive and negative spillover between work and nonwork spheres, which were excluded because of the conceptual divergence [53]; (b) only studies using statistical methods were extracted for the illustration of a corroborated relationship between balance and its antecedents and outcomes (e.g., [49,54,55]). After removing the duplicates, two reviewers (K.P.W. and P.-L.T.) independently screened the titles and abstracts of the studies in accordance with the inclusion and exclusion criteria. The full-text screening was conducted by three independent reviewers (K.P.W., P.-L.T. and A.H.S.C.), who resolved the conflicting results through discussion. Finally, a total of 99 articles were extracted. Details of the article selection process are demonstrated in Figure 2. The coverage of the journals included management, psychology, occupational health, ergonomics, education, hospitality, tourism and other fields. This range showed that the influences of WLB are pertinent to various industries where many professionals also intend to enhance WLB. Although not all empirical studies of WLB were located, because some were inaccessible in the database or because the keywords searched were not reflected in the title or abstract of the articles, sufficient articles were located to enable the identification of research gaps.

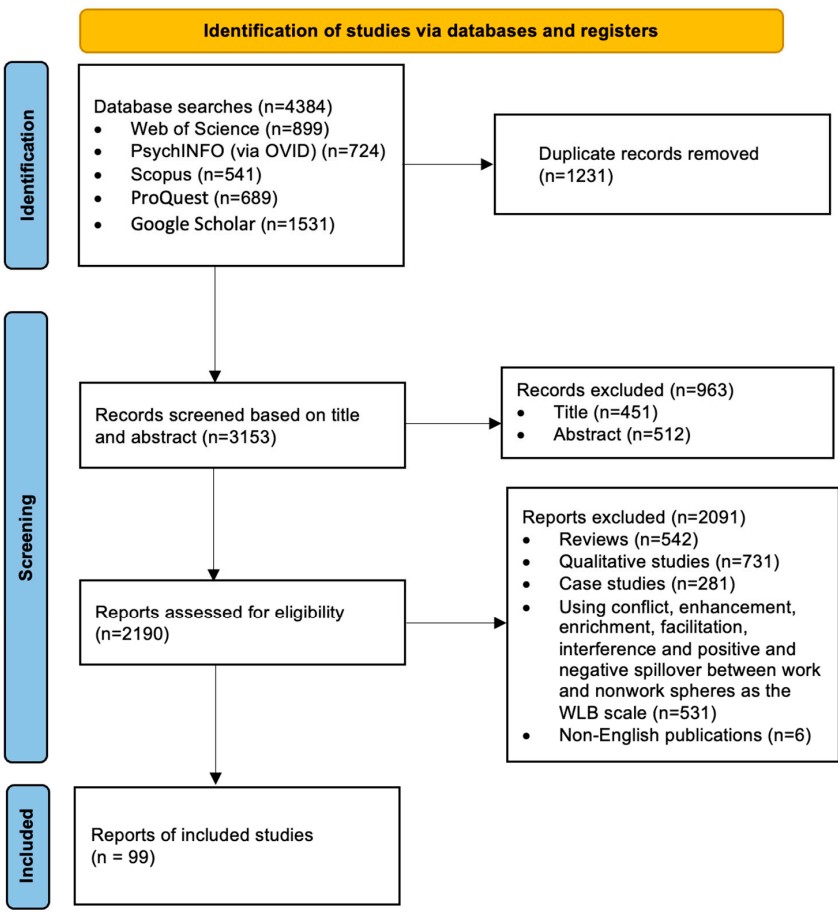

**Figure 2.** PRISMA flow-chart of the study selection process.

### 2.3. Data Coding and Analysis

The 99 articles were coded into specified parameters by two reviewers (K.P.W. and P.-L.T.): data collection methods, sources, data analysis methods, countries investigated, the pivot of the WLB scale and thematic topic classified as antecedents and outcomes of WLB. The coding of parameters aimed to assist in identifying research gaps. To facilitate the evaluation of the thematical trend, the articles were categorised into three five-year time periods: 2006–2010 ($n = 19$); 2011–2015 ($n = 39$); and 2016–2020 ($n = 41$). The increasing number of articles reflects that WLB remains an active societal issue.

## 3. Results

### 3.1. Study Characteristics

The sample sizes of the included studies ranged from 79 to 59,121. A total of 17 industries were identified from the selected studies, including administrative and support, agriculture, catering, education, engineering, environment, financial, health care, hospitality, information technology, manufacturing, pharmaceutics, public sector, service, telecommunications, transportation and retail (see Table 1). In total, 24 countries were identified from 91 studies, and a further 8 studies that stated 'Europe' as the investigated target, without stating the countries, were thus uncounted. Some studies investigated more than one country, and hence frequency was used for counting the number of the countries considered in the articles. The United States ($n = 24$) was the most popular country in the investigation of WLB, followed by Australia ($n = 11$) and India ($n = 7$). The findings showed that WLB is considered more in Western countries ($n = 69$) than in Asian countries ($n = 29$) and African countries ($n = 1$), and thus more investigations need to be conducted to consider the needs of Asian and African workers. The small number of investigations in Asian and African countries could not raise awareness of the issue of WLB in society. In fact, long working hours in Asian countries were more common than in Western countries [56]. Details of the participant characteristics are shown in Supplementary Materials Table S1.

**Table 1.** The number of industries identified in the 99 articles.

| Industry | Number |
|:---|:---:|
| Administrative and support services industry | 7 |
| Agriculture industry | 1 |
| Catering industry | 2 |
| Education Industry | 9 |
| Engineering industry | 3 |
| Environmental industry | 1 |
| Financial industry | 9 |
| Health care Industry | 16 |
| Hospitality industry | 4 |
| Information Technology Industry | 6 |
| Manufacturing Industry | 10 |
| Pharmaceutical Industry | 2 |
| Public sector | 3 |
| Service industry | 7 |
| Telecommunication industry | 1 |
| Transport Industry | 2 |
| Retail industry | 1 |

### 3.2. Study Design

All studies employed cross-sectional surveys for data collection, and only a few also used a mixed method approach (i.e., combination of interviews and survey) ($n = 7$). The studies with interviews were intended to identify the predictors of the phenomenon. These factors need to be further analysed with survey data. Most of the studies ($n = 94$) used primary datasets, and five studies used secondary datasets. More than half of the studies ($n = 52$) utilised regression analysis (including regression and correlation). In the last

10 years, confirmatory factor analysis (CFA) and structural equation modelling (SEM) were frequently used. Details of the study designs are shown in Supplementary Materials Table S1.

*3.3. Pivots of WLB Scale*

The articles adopted different WLB scales to measure WLB's relationship with various antecedents and outcomes. As the WLB scale of some articles contained one or more foci, frequencies were used to measure the pivot. The WLB scales were classified into 10 different pivots: perceived satisfaction between work and nonwork ($n = 43$); organisational strategies and practices ($n = 25$); sufficient time off ($n = 18$); influence on physical and mental states ($n = 14$); work–nonwork interference ($n = 12$); meeting role expectations ($n = 11$); accomplishment of role responsibilities ($n = 5$); role overload ($n = 5$); time management ($n = 3$); and job demand ($n = 3$) (see Table 2). "Perceived satisfaction between work and nonwork" and "Organisational strategies and practices" were mainly used for the scale at the beginning, and others were applied in the most recent 10 years. To date, "perceived satisfaction between work and nonwork" was more favoured than others. Adopting "Sufficient time off", "Influence on physical and mental states", "Work-nonwork interference" and "Meeting role expectations" was more common across the extracted studies compared to the pivots of the WLB scales, "Accomplishment of role responsibilities", "Role overload", "Time management" and "Job demand". This implied that subjective assessments of the balance between the work and personal life domains were the emphasis of defining WLB, rather than evaluations of tangible support, quantitative evaluations of the amount of time spent in different domains and objective assessments of the load and demands of each role. Differences in the frequencies of the pivots of WLB scales imply that the use of WLB scales was inconsistent across the studies, and the deviations in the results may be due to the use of different WLB scales.

**Table 2.** The frequencies of the pivots of the WLB scale used in the 99 articles.

| Pivots of WLB Scale | Frequencies | Description |
|---|---|---|
| Perceived satisfaction between work and non-work | 43 | Satisfied with the balance between work and life/ feeling successful in balancing work and family or life/being able to maintain a healthy WLB |
| Organisational strategies and practices | 25 | Policies or practices for employees to balance work and family or life responsibilities (e.g., dependent care, paternity leave, flexible working hours, etc.)/promotion of WLB policies/supportive supervisor |
| Sufficient time off | 18 | Time for family and personal life |
| Influence on physical and mental states | 14 | Feeling positive, optimistic, stressed, worried or tired in relation to work, family or personal life |
| Work–nonwork interference | 12 | The interaction between work and nonwork domains |
| Meeting role expectations | 11 | The ability to meet the role expectation of work, family, community, etc. |
| Accomplishment of role responsibilities | 5 | The ability to fulfil the responsibilities related to work, family and personal roles |
| Role overload | 5 | Experiencing long working hours/high job demands or family demands |
| Time management | 3 | Work time schedule/Time for personal life |
| Job demand | 3 | Physical and/or psychological involvement in a job |

*3.4. The Thematic Topics of the Antecedents and Outcomes of WLB*

A wide variety of antecedents and outcomes were identified from the 99 articles. Based on the natures of the antecedents and outcomes, they were categorised into eight main thematic topics: organisational behaviour, wellbeing, WLB practices, work characteristics,

family/personal context, work time/schedule, human resources management strategies and resource supports (See Table 3a–h). Some variables were more favoured than others, for instance, job satisfaction, psychological and emotional wellbeing and working hours (including long and overtime working hours). However, it was also noted that some variables became less noted in 2016–2020 amongst the empirical studies of WLB, such as work–nonwork conflict, turnover intentions, organisational commitment and flexible work time.

**Table 3.** (**a**) Antecedents and outcomes of WLB in organizational behaviour in 2006–2010, 2011–2015 and 2016–2020. (**b**) Antecedents and outcomes of WLB in wellbeing in 2006–2010, 2011–2015 and 2016–2020. (**c**) Antecedents and outcomes of WLB in WLB practices in 2006–2010, 2011–2015 and 2016–2020. (**d**) Antecedents and outcomes of WLB in work characteristics in 2006–2010, 2011–2015 and 2016–2020. (**e**) Antecedents and outcomes of WLB in family/personal context in 2006–2010, 2011–2015 and 2016–2020. (**f**) Antecedents and outcomes of WLB in work time/schedule in 2006–2010, 2011–2015 and 2016–2020. (**g**) Antecedents and outcomes of WLB in human resources management strategies in 2006–2010, 2011–2015 and 2016–2020. (**h**) Antecedents and outcomes of WLB in resources supports in 2006–2010, 2011–2015 and 2016–2020 (Note: (+) represents positive correlation with WLB; (−) represents negative correlation with WLB; * was the sum of items in the five-year time period).

| | 2006–2010 (*n* = 19) | | | | 2011–2015 (*n* = 39) | | | | 2016–2020 (*n* = 41) | | | | |
| --- | --- | --- | --- | --- | --- | --- | --- | --- | --- | --- | --- | --- | --- |
| | Antecedents | | Outcomes | | Antecedents | | Outcomes | | Antecedents | | Outcomes | | |
| | (+) | (−) | (+) | (−) | (+) | (−) | (+) | (−) | (+) | (−) | (+) | (−) | |
| **(a)** | | | | | | | | | | | | | |
| **Organisational behaviour** | | | | | | | | | | | | | Sub-total |
| Job satisfaction | | | 5 | | | | 10 | | | | 11 | | 26 |
| Organisational & management support | | | | | 6 | | | | 8 | | | | 14 |
| Work-nonwork conflict | | | | 1 | | 4 | 4 | | | 4 | | | 13 |
| Turnover intentions | | | | 2 | | | | 6 | | | | 4 | 12 |
| Work stress | | 2 | | 2 | 2 | | | | | 5 | | | 11 |
| Organisational commitment | | | 2 | | | | 5 | | | | 2 | | 9 |
| Job performance | 1 | | 1 | | | | 1 | | | | 4 | | 7 |
| Supervisor support | 1 | | | | | | | | 6 | | | | 7 |
| Job involvement | | 1 | | | | 3 | | | | 2 | | | 6 |
| Work-nonwork enrichment/facilitation | | | | | 2 | | 1 | | 3 | | | | 6 |
| Work engagement | | | | | | | 1 | | 4 | | 1 | | 6 |
| Work motivation | 2 | | | | 1 | | | | 1 | | | | 4 |
| Innovation | | | | | | | 1 | | | | 2 | | 3 |
| Organisational citizenship behaviour | 1 | | | | | | | | | | 2 | | 3 |
| Quality of products and/or services | | | | | | | 3 | | | | | | 3 |
| Cooperation | 1 | 1 | | | | | | | | | | | 2 |
| Co-worker support | | | | | | | | | 2 | | | | 2 |
| Productivity | | | | | | | 1 | | | | 1 | | 2 |
| Transformational leadership | | | | | 2 | | | | | | | | 2 |
| Work loyalty | 1 | | | | | | 1 | | | | | | 2 |
| Work permeability | | | | | 2 | | | | | | | | 2 |
| Learn and reflect | | | | | 1 | | | | | | | | 1 |
| New working system | | 1 | | | | | | | | | | | 1 |
| Organisational direction | 1 | | | | | | | | | | | | 1 |
| Organisational financial performance | | | | | | | 1 | | | | | | 1 |
| Organisational pride | | | | | | | | | | | 1 | | 1 |
| Performance appraisal | | 1 | | | | | | | | | | | 1 |
| Personal–organisation fit | | | | | 1 | | | | | | | | 1 |
| Professional commitment | | | | | | | 1 | | | | | | 1 |
| Recognition | | | | | 1 | | | | | | | | 1 |
| Role clarity | | | | | | | | | 1 | | | | 1 |
| Supervisory responsibilities | | | | | | 1 | | | | | | | 1 |
| Workplace inclusion | | | 1 | | | | | | | | | | 1 |
| Sub-total * | | | 28 | | | | 62 | | | | 64 | | 154 |

**Table 3.** *Cont.*

| | 2006–2010 (n = 19) | | | | 2011–2015 (n = 39) | | | | 2016–2020 (n = 41) | | | | |
|---|---|---|---|---|---|---|---|---|---|---|---|---|---|
| | **Antecedents** | | **Outcomes** | | **Antecedents** | | **Outcomes** | | **Antecedents** | | **Outcomes** | | |
| | (+) | (−) | (+) | (−) | (+) | (−) | (+) | (−) | (+) | (−) | (+) | (−) | |
| **(b)** | | | | | | | | | | | | | |
| **Wellbeing** | | | | | | | | | | | | | Sub-total |
| Psychological wellbeing | 1 | | 5 | | | | 6 | | 4 | | 8 | | 24 |
| Emotional wellbeing | | | 2 | | 8 | | | | 8 | | 2 | | 20 |
| Burnout | | | 1 | | | 1 | | 4 | | 3 | | 1 | 10 |
| Social wellbeing | 2 | | | | 1 | | 3 | | 4 | | | | 10 |
| Life satisfaction | | | 1 | | | | 2 | | | | 4 | | 7 |
| Stress | | | 1 | | | | | 4 | | | | 2 | 7 |
| Anxiety/Depression | | | | | | | | 2 | | | | 3 | 5 |
| Life control | | | 1 | | | | | | 2 | | | | 3 |
| Physical wellbeing | | | | | 1 | | | | 2 | | | | 3 |
| Sleep problem | | | 1 | | | | | | | 1 | | | 2 |
| Societal wellbeing | | | | | 2 | | | | | | | | 2 |
| Workplace wellbeing | | | | | | | | | 1 | | | | 1 |
| Life orientation | | 1 | | | | | | | | | | | 1 |
| Life quality | | | | | | | | | | | 1 | | 1 |
| Life meaningfulness | | | 1 | | | | | | | | | | 1 |
| Motivation | | | 1 | | | | | | | | | | 1 |
| Need fulfilment | | | 1 | | | | | | | | | | 1 |
| Neuroticism | | 1 | | | | | | | | | | | 1 |
| Passion | | | | | | | | | 1 | | | | 1 |
| Safety | | | | | | | | | 1 | | | | 1 |
| Sub-total * | | | 20 | | | | 34 | | | | 48 | | 102 |
| **(c)** | | | | | | | | | | | | | |
| **WLB practices** | | | | | | | | | | | | | Sub-total |
| Flexible worktime | 6 | | | | 2 | | | | 3 | | | | 11 |
| Organisational support of WLB | 1 | | | | 5 | | | | 1 | | | | 7 |
| Flexible work location | 3 | | | | 1 | | | | 2 | | | | 6 |
| WLB policies | 1 | | | | 2 | | | | 3 | | | | 6 |
| Work leave | | | | | 2 | | | | 4 | | | | 6 |
| Flexible work arrangement | 1 | | | | 3 | | | | 1 | | | | 5 |
| Childcare support/subsidy | 2 | | | | | | | | 2 | | | | 4 |
| Days for holidays per year | 1 | | | | | | | | 1 | | | | 2 |
| Dependent care | | | | | | | | | 1 | | | | 1 |
| Family support culture | 1 | | | | | | | | | | | | 1 |
| Job sharing | 1 | | | | | | | | | | | | 1 |
| Switching from full-time to part-time job | 1 | | | | | | | | | | | | 1 |
| Work breaks | | | | | | | | | 1 | | | | 1 |
| Sub-total * | | | 18 | | | | 15 | | | | 19 | | 52 |
| **(d)** | | | | | | | | | | | | | |
| **Work characteristic** | | | | | | | | | | | | | Sub-total |
| Job autonomy | 2 | | | | 1 | 1 | | | 9 | | | | 13 |
| Job demands | | | | | | 4 | | | | 7 | | | 11 |
| Control over work time | 2 | | | | 1 | | | | 1 | | | | 4 |
| Job meaningfulness | | | | | 1 | 1 | | | 2 | | | | 4 |
| Customer feedback | 1 | | | | | 2 | | | | | | | 3 |
| Job diversity | 3 | | | | | | | | | | | | 3 |
| Job challenge | | | | | 1 | | | | | | | | 1 |
| Job complexity | 1 | | | | | | | | | | | | 1 |
| Job feedback | 1 | | | | | | | | | | | | 1 |
| Job identity | 1 | | | | | | | | | | | | 1 |
| Job significance | 1 | | | | | | | | | | | | 1 |
| Sub-total * | | | 12 | | | | 12 | | | | 19 | | 43 |

**Table 3.** *Cont.*

| | 2006–2010 (*n* = 19) | | | | 2011–2015 (*n* = 39) | | | | 2016–2020 (*n* = 41) | | | | |
|---|---|---|---|---|---|---|---|---|---|---|---|---|---|
| | Antecedents | | Outcomes | | Antecedents | | Outcomes | | Antecedents | | Outcomes | | |
| | (+) | (−) | (+) | (−) | (+) | (−) | (+) | (−) | (+) | (−) | (+) | (−) | |
| **(e)** | | | | | | | | | | | | | |
| **Family/personal context** | | | | | | | | | | | | | Sub-total |
| Children at home | | 3 | | | 1 | | | | | 1 | | | 5 |
| Enough time for family/personal life | 1 | | | | 2 | | | | 1 | | | | 4 |
| Family satisfaction | | | 1 | | | | 2 | | 1 | | | | 4 |
| Family support | | | | | 2 | | | | 1 | | | | 3 |
| Dependent care responsibilities | | | | | 1 | | | | | 1 | | | 2 |
| Family demands | | | | | | | | | | 1 | | | 2 |
| Quality relationship with family | | | | | | | | | 2 | | | | 2 |
| Communication about work | | | | | 1 | | | | | | | | 1 |
| Family autonomy | | | | | | | | | 1 | | | | 1 |
| Family functioning | | | 1 | | | | | | | | | | 1 |
| Family performance | | | 1 | | | | | | | | | | 1 |
| Financial wellbeing | | | | | | | | | 1 | | | | 1 |
| Living with partner | | | | | 1 | | | | | | | | 1 |
| Household chores | | 1 | | | | | | | | | | | 1 |
| Sub-total * | | | | 8 | | | | 11 | | | | 12 | 31 |
| **(f)** | | | | | | | | | | | | | |
| **Work time/schedule** | | | | | | | | | | | | | Sub-total |
| Working hours | | 4 | | | | 3 | | | | 10 | | | 17 |
| Working weekends | 1 | | | | 1 | | | | 1 | | | | 3 |
| Time pressure from work | | | | | 1 | | | | 1 | | | | 2 |
| Commuting time | 1 | | | | | | | | | | | | 1 |
| "On call" hours | 1 | | | | | | | | | | | | 1 |
| Propensity to be late for work | | | | | | | | | | | 1 | | 1 |
| Propensity to leave early | | | | | | | | | | | 1 | | 1 |
| Work time intensity | | | | | 1 | | | | | | | | 1 |
| Sub-total * | | | | 7 | | | | 6 | | | | 14 | 27 |
| **(g)** | | | | | | | | | | | | | |
| **Human resources management strategies** | | | | | | | | | | | | | Sub-total |
| Job security | | | | | 2 | | | | 1 | | | | 3 |
| Career opportunities | 1 | | | | 1 | | | | | | | | 2 |
| Employee retention | | | | | 1 | | 1 | | | | | | 2 |
| Pay | | | | | 2 | | | | | | | | 2 |
| Advancement opportunity | | | | | 1 | | | | | | | | 1 |
| Economic wellbeing | | | | | | | | | 1 | | | | 1 |
| Organisational attraction | | | | | | | 1 | | | | | | 1 |
| Organisational image | | | | | 1 | | | | | | | | 1 |
| Psychological contract breach | | | | | | | | | | | 1 | | 1 |
| Sub-total * | | | | 1 | | | | 10 | | | | 3 | 14 |
| **(h)** | | | | | | | | | | | | | |
| **Resource supports** | | | | | | | | | | | | | Sub-total |
| Resources | 1 | | | | 1 | | | | | | | | 2 |
| Use of technology | | | | | | | | | 1 | 1 | | | 2 |
| Property | 1 | | | | | | | | | | | | 1 |
| Website usability | | | | | 1 | | | | | | | | 1 |
| Sub-total * | | | | 2 | | | | 2 | | | | 2 | 6 |

### 3.4.1. Trend of Thematic Topics

The number of studies on organisational behaviour was the largest, gradually increasing from 2006–2020. Advocacy for WLB undoubtedly aims to enhance the work attitudes and performances (i.e., organisational behaviour) of workers by taking their nonwork roles into account [57]. Workers who maintain a healthy WLB are more motivated to take on job responsibilities. Satisfactory balance allows workers to work with less tension and anxiety

at work and makes it easier for individuals to perform work tasks. Motivated workers are more productive and efficient. The workers approach their work with a good attitude, and it tends to permeate the entire workplace. Therefore, WLB is a self-perpetuating productivity enhancer. The capability of an organisation to gain profit mostly hinges on its workers. The introduction of WLB considerations involves resources input from an organisation, and thus the organisation must be confident of a reward before it introduces these policies [58]. Therefore, a relatively large number of studies have evaluated the associations of variables in organisational behaviour (e.g., work stress, job satisfaction, turnover intentions and organisational commitment) with WLB [7,46,59].

Apart from organisational behaviour, the studies of wellbeing were much more common than studies of work characteristics or work time/schedule, while there was a dramatic drop in the number on the topic of human resources management strategies. The increase in the number of investigations into wellbeing does not necessarily imply that the wellbeing of workers is getting better; on the contrary, their wellbeing raises the concern of organisations and researchers. The wellbeing of workers is a critical component of the sustainable development of an organisation and a healthy working environment [2]. Organisations that promote wellbeing make it easier for employees to manage stress levels while maintaining a positive and productive workplace [2,49]. Enhancing the wellbeing of workers can help them feel valued and supported at work [50,51]. It also improves employee engagement and motivates teams across the company to achieve their work goals.

The nature of a job (e.g., job autonomy, demands and meaningfulness) impacts the personal life domain, as well as life satisfaction [50]. The long-lasting phenomenon of long working hours still jeopardises the wellbeing of workers [60]. Furthermore, the importance of human resources management strategies, which are related to fundamental employment needs, seems in recent years to be overlooked by researchers. The number of studies into WLB practices rose slightly between (2011–2015) and (2016–2020), and flexible work time accounted for the highest number of those. However, the trend to investigate flexible work time is decreasing, which might be because of the repercussions caused by the vague boundary between the work and nonwork roles. Of the variables in WLB practices, the investigation into work leave saw an increasing trend amongst other policies. There were a consistent number of studies over the last 10 years on the family and personal context and on resource support. The strategies and policies initiated by the organisation can influence the attitudes and behaviours of its workers, and this requires further attention. Amongst all the thematic topics, the frequency of family and personal context was ranked sixth, and the frequency of resource support was lower than that of all the themes.

### 3.4.2. Frequently Used Variables

The variable with the highest frequency (*n* = 26), job satisfaction (see Table 3a), was defined as an outcome of WLB in all these studies and was positively correlated with WLB across all these studies. Psychological wellbeing was defined as a factor in 5 studies and as an outcome across 19 studies (see Table 3b). All of these studies found a positive relationship between psychological wellbeing and WLB. Emotional wellbeing was defined as a factor in 16 studies and as an outcome in 4 studies (see Table 3b). In total, 16 of these studies indicated a positive relationship between emotional wellbeing and WLB, and the remaining 4 studies demonstrated a negative association between emotional wellbeing and WLB. Working hours were defined as a predictor of WLB across 17 studies, and all of these studies found a negative relationship between working hours and WLB (see Table 3f).

### 3.4.3. Variable with an Inconsistent Relationship with WLB

Six variables—life satisfaction, emotional wellbeing, job autonomy, children at home, dependent care responsibilities and use of technology—were found to have both positive and negative associations with WLB. It is somewhat surprising that a negative association could exist between life satisfaction and WLB, but most studies did demonstrate a positive association. The relationship between job autonomy and WLB might be dependent upon

the personality of workers—some workers prefer to follow instructions rather than make decisions by themselves, and job autonomy may make these workers feel stressed [61]. The number of studies demonstrating a positive association between job autonomy and WLB was greater than that demonstrating a negative relationship. Only one study indicated a negative association. Regarding children at home and dependent care responsibilities, some working parents are fond of accompanying or looking after their children or family members, while others find their children to be a limitation. When it comes to children in the family, more studies found a negative association, and only one study demonstrated a positive association. The effects of use of technologies on WLB is still a controversial issue, as the study of Gravador and Teng-Calleja [62] found a positive relationship between the use of technology and WLB, while the study of Schwartz et al. [63] demonstrated its negative relationship. These issues need further scrutiny.

## 4. Discussion

This scoping review showed that organisational behaviours- and wellbeing-related antecedents and outcomes were particularly prevalent in the empirical studies examining WLB, followed by WLB practices and work characteristics. The antecedents and outcomes in the thematic topics of human resources management strategies and resources supports were less prevalent amongst the eight thematic topics of WLB. Although organisational behaviours are the most prevalent topics for the researchers to explore in relation to WLB, some hot issues could no longer be neglected, such as flexible working, use of technologies and the wellbeing of workers. Perceived satisfaction between work and non-work is the pivot of the WLB scale most used in the studies, followed by organisational strategies and practices and sufficient time off. Despite the repeated investigation of certain antecedents and outcomes of WLB, there is still room for further exploration of several aspects that may tremendously affect workers' WLB and are possibly controversial issues in society. First, although the influences of long working hours and flexible working have been discussed many times, a few constructive and effectual strategies to deal with the issues and more detailed scrutiny are needed. Second, the role of technology is a double-edged sword in maintaining the WLB of workers, and thus the potential impacts of technology on the WLB of workers are worthwhile to be explored. Third, a more positive subjective wellbeing is the pursuit amongst most workers; however, the intimate nexus between WLB and subjective wellbeing has seldom been evaluated. Therefore, the impacts of flexible working on WLB, the effects of technologies on WLB and the influence of WLB on subjective wellbeing are discussed below.

### 4.1. WLB Scales

Ten pivots of WLB scales were identified from the extracted previous empirical studies, and "Perceived satisfaction between work and nonwork" was mostly adopted as the WLB measurement scale. Scholars started to define the meaning of WLB in the 1980s. Different types of WLB have been proposed, for instance, tangible and intangible support, subjective and objective evaluation, and demand and resource allocation. We conjectured that most researchers probably believe that WLB should be a subjective evaluation of the realisation of different aspects of life that may affect an individual. A matter will not be considered an influencing factor on their own WLB if the employee does not feel that the matter may physically and psychologically affect them. For instance, if the worker is a workaholic, he or she may be satisfied with working long hours; if workers do not have children and dependents to care for, the implementation of family leave may not improve their WLB; if the workers are postpartum working mothers, the provision of a breastfeeding-friendly workplace and maternity leave may possibly improve their WLB. These examples explained that the WLB of different workers is distinct, depending on whether their needs are fulfilled, which in turn increases their satisfaction. Hence, most studies used the pivot of "Perceived satisfaction between work and nonwork" as the WLB scale.

### 4.2. Flexible Working

Numerous research has investigated the influences of flexible working hours or flexible work locations on WLB (refer to Table 3c. WLB practices). Undoubtedly, there are huge advantages of flexible working for workers [64]. Nevertheless, debates on this issue still exist. First, flexibility allows workers to resolve personal affairs (e.g., childcare), and it is believed that more female labour is attracted when flexible working is allowed [65,66]. However, pressure can be put on co-workers if some staff must frequently cope with private issues at work. To create fairness, workers may also have to work during non-working hours [67]. Second, working from home can save time and costs on commuting, which results in more personal time [68]. However, this form of working may blur the boundary between work and home, as workers can feel that they are never 'off work', which can lead to exhaustion and loss of productivity. However, manual industries, construction, manufacturing and catering cannot benefit from flexible working. Third, recording working hours and ensuring workers are working are complicated. During the COVID-19 pandemic, many organisations have been forced to allow home-working [69]. Home-working may become a trend in the future workplace. Hence, the implementation of flexible working should be considered thoroughly, based on productivity, the wellbeing of workers and commitment, prior to the introduction of flexible working practices [70].

### 4.3. Effects of Technologies on WLB

Technologies become part and parcel of people's daily lives. Workers take advantage of technologies to cope with everything else. For instance, workers can handle work tasks outside the office or manage nonwork affairs while working. Two of the extracted papers, Gravador and Teng-Calleja [62] and Schwartz et al. [63], investigated the effects of technologies on WLB and had distinctly different results (refer to Table 3h. Resource supports). Gravador and Teng-Calleja [62] found that technologies assisted in managing family tasks and nurturing relationships with family members. However, Schwartz et al. [63] indicated that technology frustrated workers. Some researchers believe that technologies serve as a means to provide a flexible working environment for workers [71,72]. However, other studies argued that technologies adversely affected WLB because the private time of workers might be invaded by work [73,74]. Effectively and correctly taking advantage of technologies can improve the WLB of workers, as their tasks become more efficient and their goals can be effectively accomplished. For instance, with the help of smartphones, tablets and video conferencing platforms, employees can participate in important work tasks or work even when they are away from the workplace. Digital tools can be used to complete manual and time-consuming job tasks. These examples indicated that the use of technologies enhances communication, collaboration, productivity and efficiency, which in turn improves WLB.

### 4.4. Subjective Wellbeing

WLB is one of the significant factors contributing to subjective wellbeing, as a growing body of research has revealed a positive relationship between subjective wellbeing and WLB (refer to Table 3b. Subjective wellbeing refers to "three distinct but often related components of wellbeing: frequent positive affect, infrequent negative affect and cognitive evaluations such as life satisfaction" [75]. Cognitive assessment is executed based on beliefs, experiences and behaviours in the mind of an individual, in which the level of subjective wellbeing is created. The process of perceived wellbeing is that fulfilling rudimentary needs is insufficient to maintain wellbeing, which sublimates to the pursuit of an ideal status. Wellbeing is probably developed by applying self-actualisation of Maslow's hierarchy of needs [76]. Self-actualisation needs refer to the realisation of personal potential (creativity and problem solving), self-fulfilment and striving for personal growth and experiences. To achieve self-actualisation, an individual needs to firstly fulfil four needs, namely, physiological needs, safety needs, social needs and esteem needs [77]. The extant WLB literature has developed similar elements in relation to the four needs in Maslow's hierarchy of

needs [78]. For example, physiological needs refer to sufficient sleep, three meals per day and enough rest breaks; safety needs refer to safeguarding workplace safety and health and job security; social needs refer to support from co-workers, supervisors and family; esteem needs refer to recognition, job autonomy and job identity. That is, the configuration of WLB can be formed by referring to these four needs, and hence the effectiveness of the WLB scale can be improved. The wellbeing scale used in the extracted articles mainly emphasised three perspectives: emotions (e.g., happy, pleased and confident) [79,80]; mental health (e.g., depression and stress level) [81]; and health functioning (e.g., headaches and stomach and sleep disturbance) [82]. To enhance the exhaustiveness of the wellbeing scale, the coverage can be broadened by including the antecedents and outcomes related to WLB in the category of wellbeing (e.g., social wellbeing, societal wellbeing and life satisfaction). Thus, the precision of the evaluation of the relationship of WLB and subjective wellbeing can be enhanced.

### 4.5. Future Agenda

The analysis of extant literature about WLB has been demonstrated above, and the research gaps in respect to study methods, countries investigated, the pivot of WLB scales used and thematic topics of the antecedents and outcomes of WLB have been identified. There is still room to enhance the empirical research to further refine the WLB assessment.

This review contributed to the theoretical understanding of the recent development trends in WLB research, in many perspectives based on its antecedents and outcomes. The findings showed that of the eight thematic topics, organisational behaviour always received the most attention by the researchers, particularly, job satisfaction. Organisational behaviour, dominated by human resources and environment (e.g., technology and organisational structure), is strongly interrelated with four main contexts, namely, caused behaviours, individual differences, human dignity and whole-person development [83]. The way the organisations treat their employees, for instance, respecting them, recognising their diversity and showing concern for their wellbeing, may influence the satisfaction level and physical and emotional conditions, and in turn, the perceived WLB. A potential research area for further investigation is how the organisations perform in the perspectives of caused behaviours, individual differences, human dignity and whole-person development, in which the perceived WLB of employees will be affected.

An increasing emphasis on the wellbeing of workers was found in this review of the literature. Enhancing the wellbeing of workers is the cornerstone of the whole organisation, in which promoting WLB is a kind of strategy to improve the wellbeing of the workers. Psychological wellbeing seems to be one of the significant factors affecting the health of the workers. Emre and De Spiegeleare [84] stated that the issue of wellbeing of workers must be seriously addressed, or an organisation might incur long-term detrimental effects in its effectiveness and productivity. The findings indicated that the investigation into wellbeing focuses mainly on psychological and emotional wellbeing. However, apart from these two types of wellbeing, other aspects critical to personal development were encompassed, such as economic wellbeing, social wellbeing, life satisfaction and engagement. Hence, a future study is proposed to examine the effectuality of the impacts of WLB on different aspects of wellbeing.

There is a further issue worthy of our attention, and that is the adoption of WLB practices. This review found that the number of evaluations of WLB in Western countries was relatively higher than in Asian countries, whereas work–life imbalance in Asian countries is more recurrent than in Western countries [85]. In fact, WLB practices are widely implemented in most Western countries, while Asian countries put less effort into safeguarding the wellbeing of workers in comparison with Western countries [86]. Therefore, more studies are recommended to investigate whether Asian countries have implemented WLB initiatives and how workers have perceived the usefulness of these measures. Ultimately, the environment surrounding the workers is an important factor affecting their wellbeing, according to the principle developed by Olivetti [87,88]. The wellbeing of workers had an

interdependent relationship with WLB. Therefore, it is recommended that the impacts of green space on workers' WLB can be further investigated.

This review found that the pivot of "Perceived satisfaction between work and non-work" was adopted as the WLB scale amongst most studies. Therefore, future studies are recommended to use this pivot of WLB as the scale to assess its interrelationships amongst the antecedents and outcomes. In future research, a WLB scale using the concept of satisfaction between work and nonwork could be developed and validated to apply a reliable and consistent WLB scale to enhance the significance of practice and research.

### 4.6. Practical Implications

The findings of this study suggest several courses of action for governments and organisations. Firstly, considering the wellbeing of workers, it is suggested that government officials work with the leaders in different industries to re-evaluate family-related leave policies, such as paternity leave and paid maternity leave. Secondly, in an endeavour to provide a flexible working environment for workers, especially working parents, interventions are recommended to offer to all types of enterprises the opportunity to design tailormade solutions to assist workers to minimise the conflict between work and family affairs. Thirdly, working from home has become a trend in most companies due to the COVID-19 pandemic. In the long term, organisations might consider how to optimise this approach to ensure that workers are able to complete their job duties, as well as provide flexibility to some extent. Lastly, long working hours are a controversial issue in society at all times, as long working hours adversely affected the balance and wellbeing of workers [54]. Although many regions have implemented standard working hours, many workers still suffer from the side effects of working long hours. Government agencies need to better monitor companies to ensure that this does not occur.

### 4.7. Limitations

Admittedly, this study is constrained by four research limitations. First, a quality assessment of the included studies was not conducted; however, quality assessment, conventionally, is not needed to be conducted in scoping reviews [89,90]. Second, although commonly used electronic databases were adopted for paper searches, this review might not cover all empirical papers related to WLB; human error may have occurred during the paper screening process, and the coverage for keywords used for searching could be incomplete. However, the large number of papers enhances the reliability of the findings. Third, only English publications were included in this review. Some related papers published in other languages might satisfy the selection criteria. In future analysis, publications in other languages are recommended to be included. Last, most of the results of the studies extracted were assessed based on the subjective feedback of participants. Thus, some perspectives, such as health-related issues, could be examined using an objective approach (e.g., health examination) instead of a questionnaire.

### 5. Conclusions

This scoping review, covering fifteen years in public health, psychology and management journals, synthesised 99 empirical studies on the state of knowledge of WLB. It revealed the research methods, analysis methods, countries investigated, the pivot of WLB scales and thematic topics. Eight thematic topics were classified from the antecedents and outcomes of WLB. It stated the trends—the rising concern of the effects of WLB on subjective wellbeing, the impacts of technologies on the needs of workers, the implementation of flexible working for coping with the increase in working from home, the enhancement of supervision of the execution of standard working hours and the unification of the WLB scale used—that require being taken into consideration in future exploration. A healthy WLB can help employees reduce stress, prevent burnout in their professional and personal lives, and improve mental wellbeing. Organisations have to enhance the promotion of WLB and formulate relevant strategies to assist workers in developing a sustainable WLB.

The emergence of rapid technological development is changing the work mode to remote work. Instant messaging, email and social media, once confined to the office, now are flooding into workers' homes. Therefore, organisations need to effectively take advantage of these technologies to enhance the flexibility and permeability of the workers' work and life domains instead of harming their personal life. This study also found that human resources management strategies are topics being overlooked in contributing to WLB, so future research needs to strengthen the evaluation of the effectiveness of human resource management strategies to better understand the role of these strategies in improving workers' WLB. Regarding the methodological recommendation, SEM should be adopted as the analysis method instead of regression to enhance the reliability of the model measurement.

**Supplementary Materials:** The following supporting information can be downloaded at: https://www.mdpi.com/article/10.3390/su15042875/s1, Figure S1: Number of publications from 2006–2020; Table S1: Study characteristics of the included studies [91–162].

**Author Contributions:** Conceptualization, K.P.W.; methodology, K.P.W., P.-L.T. and A.H.S.C.; software, K.P.W.; validation, K.P.W., P.-L.T. and A.H.S.C.; formal analysis, K.P.W.; investigation, K.P.W.; resources, K.P.W., P.-L.T. and A.H.S.C.; data curation, K.P.W.; writing—original draft preparation, K.P.W.; writing—review and editing, K.P.W., P.-L.T. and A.H.S.C.; visualization, K.P.W.; supervision, P.-L.T. and A.H.S.C.; project administration, K.P.W. All authors have read and agreed to the published version of the manuscript.

**Funding:** This research received no external funding.

**Institutional Review Board Statement:** Not applicable.

**Informed Consent Statement:** Not applicable.

**Data Availability Statement:** Not applicable.

**Conflicts of Interest:** The authors declare no conflict of interest.

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
