# Peer review of "Seeing the Forest and the Trees: A Scoping Review of Empirical Research on Work-Life Balance"

_sustainability, doi:10.3390/su15042875_

Round 1

Reviewer 1 Report

The paper is well written and methodologically correct. Some more information should be given on any spatial aspects that affect Work-Life Balance, for example green spaces, nursery support in the workplace. On this aspect, perhaps, one could explore Adriano Olivetti's theories on the well-being of workers. this reading could lead to further inputs.

Reviewer 2 Report

The manuscript is methodologically sound, and insightful. The systematic literature review adopts the PICO framework and uses the Prisma flowchart.

Results are presented, processed and discussed both qualitatively and quantitatively. Implications are clear and concise. 

Some minor suggestions for improvement are provided below.

1. PICO stands for Population, Intervention, Comparison, and Outcome, and is a specialized framework used by most researchers to formulate a research question and to facilitate literature review. 

1a. PICO framework should be supported by relevant references.

1b. The third dimension of PICO (Comparison) is missing.

2. The text in lines 361-355 needs some corrections as indicated:

For example, physiological needs, physiological needs refer to sufficient sleep, three meals per day and enough rest breaks; safety needs refers to safeguard of workplace safety and health and job security; social needs refers to support from co-workers, supervisors and family; esteem needs refers to the recognition, job autonomy and job identity.

3. In the Limitations section the text that follows needs revision:

"Admittedly, this study is constrained by three research limitations. Several potential limitations need to be acknowledged."

More specifically, the two sentences in the above mentioned text look alike. They should be probably merged into a single sentence.

Reviewer 3 Report

Referee report on ” Seeing the Forest and the Trees: A Scoping Review of Empirical Research on Work-Life Balance”

This scoping review extracted ninety-nine published articles from 77 journals over the period of 2006-2020. The review aimed at identifying (a) the commonly used WLB scales, (b) antecedents and outcomes related to WLB and (c) frequency of the emergence of these antecedents and outcomes. The topic of the review is interesting and important. However, making a review is a difficult task, because one has to choose what to present in more detail and which points to highlight. I think this review has potential, but I’m wondering if you really are “seeing the trees”. In general, instead of lists of themes and variables, I would like to see much more examples and more detailed explanation of the findings, especially in the presentation of the antecedents and outcomes. As a technical detail, Table 2 needs to be divided into several smaller tables, in the current format it is very hard to read.

Specific comments (in no particular order)

Line 16- “This review aims to provide an overview of empirical studies investigating the antecedents and outcomes of WLB.” As I indicate later in my comment, I don’t think “antecedents” is the best possible word.

Line 18- “The research methods, analysis methods, countries investigated, the pivot of WLB scales used, and thematic topics were identified from each article in which insufficiency was addressed.” This is very unclear, I don’t understand this sentence. I understand this: “The research methods, analysis methods, countries investigated, the pivot of WLB scales used, and thematic topics were identified from each article.” But I don’t understand what do you mean by:  “…..in which insufficiency was addressed.” Do you mean that after identifying what has been studied, you evaluated what are the areas that need more research?

Line 62- “In this scoping review, the trends and development of WLB have been updated through previous empirical studies by exploring the commonly used WLB scales that focused only on the cause-and-effect related to role balance, determining the pivots of the WLB scales, distinguishing the antecedents and outcomes of WLB and identifying the main thematic topics from the antecedents and outcomes of WLB.” This sentence is too long and therefore unclear.  

Line 86- “Role balance theory, which suggested that workers sought meaningful experiences in both work and family roles, was proposed [26].” Is there something missing from this sentence. “…, was proposed by ?”

Line 115. “The antecedents of WLB such as working hours and organisational support continue to expand in line with the WLB policies formulated by organisations and the impacts of technology [46-48]”. Throughout the paper it remains somewhat unclear to me what you mean by “antecedents”. Would “factors underlying WLB” or “predictors of WLB” be better terms in this context?

I also think that you need to explain more and give some examples of how and why, for example, working hours or organizational support are exactly connected with WLB. Now you don’t even say the direction of this connection. For example: The more a person gets support from the organization, the easier it is to achieve better WLB. Organisational support can manifest itself as XX XX XX. I miss more examples like this.

Line 117- “The perceived outcomes of WLB are also expanding as the influences focus not only on satisfaction with life and work; feelings and performances at different roles, and physical and mental health; but on creativity and financial wellbeing as well.” Similarly as with antecedents, I think more explanation and more detailed examples of these outcomes of WLB are also needed. How and why is WLB connected with, for example, satisfaction with life and work, or with physical and mental health?

Line 196- “..thus more investigations need to be conducted to consider the needs of Asian workers.” What about African workers?

Line 197- “The small number of investigations in Asian countries could not raise awareness of the issue of WLB in society.” Do you mean that the small number of investigation may not raise awareness…” And once again, what about African countries?

Line 203. “The studies with interviews were intended to identify the predictors of the phenomenon which were intended to be further analysed in empirical studies.” This sentence is very unclear, you need to clarify what you mean by this.  “The studies with interviews were intended to identify the predictors of the phenomenon. These predictors were further analysed with survey data? Or what?” “Empirical studies” is not clear enough.

“3.3 Pivots of the WLB scale”. This chapter is very short. I would like to see more discussion on these measures of WLB.

Table 1. I think the definition of “Accomplishment of role responsibilities” is incorrect in Table 1. It should be defined as “The ability to fulfil the responsibilities related to work, family, and personal roles”. Similarly, “Role overload” should be defined as “Experiencing long working hours/high job demands or family demands”.

I also don’t quite understand how “Job demand” means “Receiving support from family”. This needs more explanation in the text.

Line 221- “This finding implies that the use of WLB scales was inconsistent amongst the studies in which deviations of the results were caused by using different WLB scales.” I don’ understand this sentence. What finding implies what? What do you mean by: “the use of WLB scales was inconsistent amongst the studies in which deviations of the results were caused by using different WLB scales”. Do you mean that the studies that found different results used different WLB scales, or what do you mean?

“3.4. The thematic topics of the antecedents and outcomes of WLB”. In general, I would like to see more examples and explanation in this chapter, not just lists of different variables or themes.

Table 2 is very difficult to read. You should divide Table 2 into several smaller tables, for example according to the main themes, like organizational behaviour separately, wellbeing separately etc.

Line 230- “Some variables were more favoured than others, for instance: job satisfaction, psychological and emotional wellbeing, job autonomy and working hours (including long and overtime working hours)” Job demands is almost as common as job autonomy.

Here it would be good to explain the findings related to these variables in more detail. It should be mentioned that in every study, job satisfaction is used as outcome of WLB. It would also be good to explain what (+) in this case means. Is the finding in all studies that higher WLB increases job satisfaction? This way it would be easier for a reader to interpret the results in Table 2.

Likewise, it would be good to mention that psychological and emotional wellbeing are used both as “predictors” and outcomes of WLB. Positive connection is found for psychological wellbeing in all cases. This means that better psychological wellbeing results in higher WLB, and that higher WLB results in better psychological wellbeing. When emotional wellbeing is used as “predictor” of WLB, a positive connection is always found. Better emotional wellbeing increases WLB. However, two studies that use emotional wellbeing as an outcome of WLB find a negative connection. This means that in these studies it has been found that higher WLB decreases emotional wellbeing. On the other hand, there are also two studies that show an opposite result. Job autonomy and working hours are in all reviewed studies used as a “predictor” of WLB.  

Line 232- “However, it was also noted that over time some variables became less noted amongst the empirical studies of WLB, such as work-nonwork conflict, turnover intentions, organisational commitment and flexible work time. Can you really make a conclusion like this? It seems that work-nonwork conflict, turnover intentions, organisational commitment were more common in 2011-2015 than in 2006-2010, but then decreased somewhat in 2016-2020. This is not the same as “over time became less noted”.

Line 242- “Thus, a relatively large number of studies have evaluated the associations of variables in organisational behaviour (e.g. job satisfaction, turnover intentions and organisational commitment) with WLB.” I think work stress is also worth mention here.

Line 245- “…the number of studies of wellbeing, work characteristics and work time/schedule increased considerably over the fifteen-year period.” Perhaps it would be good to mention that of these, studies of wellbeing were much more common than studies of work characteristics or work time/schelude.

Line 248- “The increase in the number of investigations into wellbeing does not necessarily imply that the wellbeing of workers is getting better: on the contrary, their well-being may be threatened, which raises the concern of organisations and researchers.” A couple of examples supporting this claim is needed. What is investigated and how does this imply that the well-being of workers is threatened?

Line 259- “Of the variables in WLB policies, the investigation into work leave saw an increasing trend amongst other policies.” I think you mean variables in WLB practices, not WLB policies.

Line 260-262. “There were a consistent number of studies over the last ten years into the family and personal context and into resource support.” Maybe it would be good to mention how does the number of studies of these issues compare with each other and with other areas of research?

Line 268- “The relationship between job autonomy and WLB might be dependent upon the personality of workers – some workers prefer to follow instructions rather than make decisions by themselves and job autonomy may make these workers feel stressed.” You need to mention that there were much more positive than negative, only one study found a negative association.

Line 271- “Regarding children at home and dependent care responsibilities, some working parents are fond of accompanying or looking after their children or family members, while others find their children to be a limitation. Thus, two opposite results were found.” It needs to be mentioned that when it comes children at home, there were more negative associations, only one positive. Dependent care responsibilities, one positive and one negative.

Line 273- “The effects of use of technologies on WLB is still a controversial issue as many studies indicated that using technologies (e.g. mobile phone) to deal with work in nonworking time generates detrimental effects; however, some argued that technologies, for example, permitting coping with urgent personal affairs during work, provide flexibility. These issues need further scrutiny.” Since you conclude like this, I think you should mention that there were only 2 studies dealing with this issue, and one found a negative and one a positive association between use of technology and WLB.

Line 291- “First, although the influences of long working hours and flexible working have been discussed many times, few constructive and effectual strategies to deal with the issues and more detailed scrutiny are needed. Second, the role of technology is a double-edged sword in maintaining the WLB of workers and thus the potential impacts of technology on the WLB of workers are worthwhile to be explored.” Why do you talk about flexible working and impact of technology under the heading 4.1, but then discuss about these separately under the headings 4.2 and 4.3?

Line 296- “Third, a more positive subjective wellbeing is the pursuit among most workers; however, the intimate nexus between WLB and subjective wellbeing has seldom been evaluated.” What do you mean by “subjective wellbeing”? And , when you say that it has seldom been evaluated, do you at the same time imply that psychological and emotional wellbeing, that are investigated a lot, do not reflect subjective wellbeing? And why is this topic mentioned under the heading 4.1, since it is discussed separately under the heading 4.4?

Line 319- “Two of the extracted papers, Gravador and Teng-Calleja [58] and Schwartz et al. [68], investigated the effects of technologies on WLB and make very different findings on the relationship between the use of technologies and role balance.” It remains unclear what is investigated in these papers and what are the findings, so this needs more explanation.

Line 357- “To achieve self-actualisation, an individual needs to first fulfil four needs, namely, physiological needs, safety needs, social needs and esteem needs, must firstly be met [82].”  Needs to first fulfil and must firstly be met is unnecessary repetition.

Line 361- “For example, physiological needs, physiological needs refer to sufficient sleep, three meals per day and…” Physiological need is said twice.

Lin 413- “The findings of these studies suggest several courses of action for governments and organisations.” The findings of what studies? The findings of this study?

line 423- “Lastly, long working hours are a controversial issue at all times that the balance and wellbeing of workers are adversely affected.” I don’t understand this sentence. Do you mean that long working hours are a controversial issue? And do you mean that long working hours affect the balance and wellbeing of workers adversely? Or do you mean that findings regarding how long working hours affect the balance and wellbeing of workers differ from each other?

Line 429- “Admittedly, this study is constrained by three research limitations. Several potential limitations need to be acknowledged.” First you say that there are three limitations. Then you repeat and say there are several potential limitations. After that you list three limitations and one “last” limitation. This is very unclear.

Reviewer 4 Report

Thank you for the opportunity to the review the paper titled “Seeing the Forest and the Trees: A scoping review of empirical research on work-life balance”. This was an interesting read.

I would like to suggest some changes which might help to focus the paper more.

Introduction

Introduction might need reworking to clearly explain the motivation for the study and the current gap, which this study addresses. My main suggestion would be to justify the need for the study more. It is not currently clear why the focus is on the role balance, how the role balance is specific and different from the WLB. It would be good to better describe the relationship between the role balance and the WLB as well.

The authors claim that there are previous Literature reviews on the topic, however it is not clear how this one is different and what it contributes to the knowledge. Again, this perhaps comes to the question why it is important to focus on the role balance and what it is.

First, the role balance needs to be clearly defined in the first part of the introduction. There is a need to explain how it is different from the conflict or enrichment and perhaps provide an example of a role balance. It is also necessary to explain why it is important to look at the role balance specifically. Authors need to make clear whether they talk about WLB or role balance, as WLB is a broader concept and as the authors show in their figure includes enrichment, conflict, balance of responsibilities, etc.

Second, in the second part of the Introduction various theories related to WLB are discussed. How are they related to the role balance and explain the role balance? Overall, the focus on the role balance in the second part is rather blurred.

Third, the research aims and questions are not specific to the role balance, but rather broadly look at the WLB.

The figure 1 does not really present the essence of role balance.

I would also suggest to proof read the Introduction making sure that all the statements are clear. For example, the first sentence is too long and needs to be split.

“However, many researchers have argued that the family does not dominate the nonwork domain and that neither positive nor negative effects were generated by each role in life [2,6,7]. Thus, WLB has been suggested as the solution to this problem and has become the primary focus in society. The term balance here means he stability of the physical body and mindset [8].” – not clear what this means and whether the authors are going to use the term balance in the same sense.

Methodology

There is a need to explain the choice of databases for the review, as the authors acknowledge in the Limitations that possibly more literature on the topic exists elsewhere.

It is not clear what is meant by the period 1st January – 31st March. Does it mean that the papers were downloaded within this period? If yes, there is a need to clearly say so. What was the period for which the papers were reviewed – this needs to be stated in years in the inclusion criteria (I assume 2006-2020). It might make sense to present a couple of graphs in Supplementary files perhaps showing the trends over years in terms of publications, also maybe with the Journals which focus more on this topic.

As PICO framework was used – what was the comparison?

How the antecedents are covered by the search string is not clear as the list is mainly for outcomes.

It would be useful perhaps to add an exemplar paragraph from one of the papers to illustrate the role balance. This will help the readers to understand how the papers on role balance were identified and selected in comparison with enrichment and conflict.

Results

The sub-title ‘Participant characteristics’ does not seem to reflect the content of the section.

This study is not a meta-analysis, so it is not clear why to report the total sample size.

The list of industries could be provided in the table instead of in-text with the number of papers for each industry.

Given that the selected studies were about the role balance, the conclusion about which countries focus on WLB might not be accurate as studies from different countries might study different aspects of WLB (e.g. role conflict, enrichment).  

Not clear what is meant by ‘pivot’.

Lines 221-223: “This finding implies that the use of WLB scales was inconsistent amongst the studies in which deviations of the results were caused by using different WLB scales.” – not clear what this means.

I was wondering which scales were specific to the role balance. How actually the role balance is measured in the literature. It might be good to say a few words about that.

It would be good to make clear whether the authors talk about the antecedents and outcomes of WLB or role balance.

I would suggest to overall reorganise the section 3.4. and present the findings in a more coherent manner. Maybe first speak about the main clusters of antecedents and then about the main clusters of outcomes. They may be organised at different levels for example (individual, organisational, environmental) and then presented in accordance with the clusters in Table 2. Further subtitles may be of use here.

I struggled to see in the findings the fourth aim of the study was addressed – enumerate (not clear what enumerate means here) the correlations among WLB, antecedents and outcomes for the periods. This might require a separate section in the Results to identify the key trends.

 Discussion

 I would suggest the change the subtitle of the first section of the Discussion (4.1.) as it presents the overview or summary of the findings.

The discussion currently focuses on the under researched areas. It seems to me that it would be good to discuss also the findings related to the well-researched areas and present what we know about the role balance. The discussion of the under researched areas could be combined with the directions for future research. It would also be good to discuss the findings related to the used measurement scales as their evaluation was one of the aims of the study.

Lines 325 – 343: Discussion of the impact of Social media – this does not seem to be based on the Findings and looks more like some speculations. I would suggest to make sure that the section on technologies is more based on the findings from the literature review discussing what we know from the analysed literature and what we need to learn.

As suggested above it might make sense to combine the discussion of the under researched areas with the suggestions for future studies.

It is also not clear how the practice and policy implications are based on the research findings. This connection needs to be made clearer. Perhaps some restructuring of the Results and Discussion sections with the clear emphasis on what we already know will help here.

Thank you for the opportunity to review this manuscript and I wish the authors god luck in their revisions.

Round 2

Reviewer 4 Report

 Thank you for the opportunity to the review the revised version of the paper titled “Seeing the Forest and the Trees: A scoping review of empirical research on work-life balance”. I would like to thank the authors for addressing the previous comments.

Below are some further suggestions:

1.       I think the focus on antecedents as well as outcomes was good and did not see the point of changing the word ‘antecedents’ to the word ‘factors’. If the authors prefer the word factors, it is necessary perhaps to explain what is meant by factors.

2.       The first sentence in the Introduction is a bit clunky and will benefit from re-writing

3.       Line 36 – I was wondering if it would be better to replace rapport with relationship.

4.       It might be better to use the word foci instead of focuses (pl)

5.       Sentence “However, many researchers have argued that the family  does not dominate the nonwork domain” needs a reference.

6.       “Thus, WLB has been suggested as the solution to this problem and has become the primary focus in society” – not clear to which problem, the authors might need to state this problem clearer in the previous sentences, since the focus on WLB is the key novelty of this study.

7.       “It is because balance is conceptually different from conflict. Conflict and enrichment include  one domain's characteristics that affect or shape another domain, while balance has not  yet emphasised it. The emergence of conflict is related to the expected behaviours of employees; nevertheless, WLB is to manage the conflicts between work and life roles which 56 is conceptually more demanding than conflict. Fig. 1 illustrates how conflict and enrichment exist in the state of WLB. The overlap between work and life domains indicated  conflict and enrichment. WLB is a dynamic state and the interaction between work and life domains causes conflict or enrichment. The conflict or enrichment may vary and then create a new state of balance.” - These are very important sentences that unpack the motivation for the study and they need re-writing for clarity. Also, the concept of the WLB is not clear here. The authors might need to cite a well-accepted definition of the WLB, which underscores its specificity and difference from conflict and enrichment. This definition will also help to understand how the studies were selected for the review.

8.       It is also important to clearly state in the Introduction the motivation for this study (please see my comments to the previous version). Why is it important to review the WLB literature? We understand that WLB is different from conflict and enrichment, but why is it important to look at balance rather than conflict or enrichment? In other words, why is this study necessary? What will it add to our knowledge? This is important to state given the previous literature reviews on the topic.

I would overall recommend to have a look at the literature review by Hillman and Guenther (2021) for structuring of the paper, articulation of motivation and contributions.

Hillmann, J., & Guenther, E. (2021). Organizational resilience: a valuable construct for management research?. International Journal of Management Reviews, 23(1), 7-44.

9.       I actually liked the focus on the scales development. If the authors argue that WLB is conceptually different from conflict and enrichment, it is necessary to understand how it is measured and whether the existing scales capture this conceptual difference. I think this contribution can be emphasised by the authors more.

10.    History and theory of WLB – this section might benefit from articulation of the main areas which require further investigation. Again, this section should clarify the motivation and intended contribution of this study.

11.   I would like to thank the authors for adding sub-headings to the Findings section

12.   Line 296: “seems in recent years to be being overlooked by researchers.” – delete ‘being’.

13.   Discussion - I would still suggest to extend the summary of the findings in the first paragraph (add more about what we already know) – it will make much easier to follow the identified gaps and future research suggestions.

14.   WLB scales – I would like to thank the authors for the summary of the findings. I was wondering if some pivots are missing in the scales given the definition of WLB the authors use.

15.   It might be useful to clarify what sections 4.2- 4.4 discuss. Do they discuss the key findings, key trends or key gaps? How are these sections related to Table 3 (summary of findings)? I would suggest to explain this relationship. For example, have the authors clustered the findings in these three categories? Which findings go under which category? Have the authors identified the emerging trends? Are these sections discussing variables with controversial findings?

16.   Also, it would be good to see how the Future research section is connected to the Discussion.

17.   I would also expect a more extended discussion of the WLB scales and their change for the needs of the future research. 

18.   Same for implications – don’t see well their connection to the Discussion section.

19.   I was surprised to see some further recommendations in the Conclusion, which were not discussed in detail before (Longitudinal research and HRM practices).

Thank you for the opportunity to review this manuscript and I wish the authors good luck.

Author Response

Please see the attachment. Thank you very much for your comments and suggestions.
